# T Cell Immunity to Bacterial Pathogens: Mechanisms of Immune Control and Bacterial Evasion

**DOI:** 10.3390/ijms21176144

**Published:** 2020-08-26

**Authors:** Freya R. Shepherd, James E. McLaren

**Affiliations:** Division of Infection and Immunity, Cardiff University School of Medicine, Cardiff CF14 4XN, UK; shepherdf1@cardiff.ac.uk

**Keywords:** T cell immunity, major histocompatibility complex-restricted T cells, αβ T cells, γδ T cells, MAIT cells, Human Leukocyte Antigen alleles, bacterial infection, immune evasion, virulence, superantigens

## Abstract

The human body frequently encounters harmful bacterial pathogens and employs immune defense mechanisms designed to counteract such pathogenic assault. In the adaptive immune system, major histocompatibility complex (MHC)-restricted αβ T cells, along with unconventional αβ or γδ T cells, respond to bacterial antigens to orchestrate persisting protective immune responses and generate immunological memory. Research in the past ten years accelerated our knowledge of how T cells recognize bacterial antigens and how many bacterial species have evolved mechanisms to evade host antimicrobial immune responses. Such escape mechanisms act to corrupt the crosstalk between innate and adaptive immunity, potentially tipping the balance of host immune responses toward pathological rather than protective. This review examines the latest developments in our knowledge of how T cell immunity responds to bacterial pathogens and evaluates some of the mechanisms that pathogenic bacteria use to evade such T cell immunosurveillance, to promote virulence and survival in the host.

## 1. Introduction

The human body has evolved a complex symbiotic relationship with innocuous, commensal bacteria in which immunological defense mechanisms are tempered, to allow a state of mutualism that is beneficial to both the host and the colonizing microbiota. However, mucosal surfaces of the human body also frequently encounter harmful bacterial pathogens and, in response, employ immune defense mechanisms designed to counteract such pathogenic assault. Recognition of pathogen-associated molecular patterns on the surface of bacteria, such as lipopolysaccharides and endotoxins, initially triggers a rapid, first-line of defense that is driven by the innate immune system. Such protective immunity was originally thought to not be retained in any form of enduring immunological memory. However, this concept was recently challenged, revealing that cellular components of the innate immune system possess “trained immunity” and retain the ability to respond faster, upon pathogen reencounter [1]. Immunological memory is more classically witnessed in the adaptive immune system, composed of T cells and B cells, which respond to foreign antigens from pathogens to orchestrate protective immune responses that persist and are memorized. In the T cell arm of adaptive immunity, CD8^+^ T cells function to kill infected cells, whilst CD4^+^ T cells support memory CD8^+^ T cell responses [2,3] and antibody-generating B cells [4], whilst also possessing direct killing capabilities. T cells use membrane-bound, heterodimeric T cell receptors (TCRs) to detect and respond to infected cells expressing foreign antigen, and are classically sub-divided based on their expression of αβ or γδ TCRs. Conventional αβ T cells typically recognise small peptide antigens (8–13 amino acids) or larger antigenic fragments (>15 amino acids) presented on the cell surface of antigen-presenting cells (APCs), through major histocompatibility complex (MHC) class I or class II molecules. CD8^+^ T cells typically recognise MHC class I-restricted peptides, whilst CD4^+^ T cells detect MHC class II-restricted peptides, although it is evident that CD8^+^ T cells can break these “rules” and bind peptide-MHC (pMHC) class II complexes [5,6]. Classical MHC class I and II molecules are extremely polymorphic (>6000 allelic variants) and can present an immensely diverse array of antigens from pathogens. To respond to such antigenic variance, TCR sequences are also very diverse and TCR α- and β-chain rearrangements can theoretically generate <10^21^ distinct αβ TCRs [7], although realistically, in humans this is closer to 10^8^ TCRs [8]. To create such clonotypic diversity, somatic recombination of variable (V), diversity (D), and joining (J) gene segments (D regions are only found in TCR β-chains), and junctional adaptations, occur to produce three complementarity determining region (CDR) loops, in each TCR chain that govern recognition of the pMHC complexes. CDR1 and CDR2 loops, germline encoded from 47 TCR α-chain V (TRAV) and 54 TCR β-chain V (TRBV) genes, typically bind the TCR to its target MHC, whilst hypervariable CDR3 loops, defined by unique nucleotide sequences, engage the exposed regions of the MHC bound peptide [7,9]. Such high diversity in the naïve T cell compartment means that the precursor frequency of antigen-specific T cells is very low [10]. Accordingly, mechanisms of clonal expansion are employed to amplify the numbers of antigen-specific effector T cells. Upon antigen exposure, naïve αβ T cells are primed by APCs to clonally expand into effector T cells in a process that is guided by immunological cues delivered by antigen [11], cytokines, and co-stimulatory interactions [12,13]. Antigen clearance leads to effector T cell contraction and conversion into memory T cells [12] that populate the blood, lymphoid organs, and mucosal tissues. Such immunological memory is maintained in the absence of antigen [14] by cytokines, such as IL-7 and IL-15 [15,16], rather than those integral to effector T cell development [17,18,19,20]. Upon re-exposure to antigen, memory T cells can then be rapidly recalled, and evoke immune responses with a greater efficacy.

Conventional αβ T cells were long considered to be only capable of recognising peptidic antigens complexed to classical MHC class I and II molecules. However, it was established that αβ T cells could engage antigens presented by non-classical MHC class Ib molecules. In humans, these include Human Leukocyte Antigen-E (HLA-E) molecules, which possess near allelic monomorphism, in contrast to MHC class I and II [21]. Additionally, αβ T cells can also respond to structurally different antigens (e.g., lipids, metabolites) presented by non-polymorphic MHC-like molecules, such as MHC class I-related gene protein (MR1) and CD1 glycoproteins [22,23,24,25]. Such “unconventional” T cell populations (Figure 1), including mucosal-associated invariant T (MAIT) cells and invariant natural killer T (iNKT) cells, appear to be “innate-like”, as they are poised to respond faster than T cells restricted by classical MHC molecules and display invariance in their TCR repertoire, which limits their clonotypic diversity [24]. Furthermore, these unconventional T cells also include T cells bearing γδ TCRs, which mediate antigen recognition in an MHC-unrestricted process that, to this date, is still poorly understood, since the identity of γδ TCR antigens is largely unclear [26]. However, recent evidence in the past decade provided some clarity and suggests that it involves butyrophilin (BTN) or BTN-like (BTNL) molecules, which are related to the B7 co-stimulatory molecules, CD80 and CD86 [26,27,28]. Furthermore, antigen recognition by γδ T cells also appears to be quite promiscuous, since they can also engage MR1 [29], CD1 variants [30,31,32,33], MHC class I homologs (e.g., MICA [34]) and pMHC class I complexes [35]. γδ T cells, like other unconventional T cell subsets such as MAIT cells, are known to facilitate immunological protection ag ainst Gram-negative and Gram-positive bacterial species [36] and respond to a myriad of structurally different antigens (e.g., lipids, metabolites) generated by bacteria. 

Research in the past ten years accelerated our knowledge of unconventional αβ or γδ T cells and how they recognize bacterial antigens and also how conventional αβ T cells provide anti-bacterial protection. However, many species of bacteria, most likely more than we are currently aware of, have evolved ways to evade host antimicrobial immune responses. Such escape mechanisms act to corrupt the crosstalk between innate and adaptive immunity, thus, potentially tipping the balance of host responses toward pathological rather than protective. Some pathogens cause chronic infections that are difficult to treat and employ virulence mechanisms that might dampen, or in some cases, also inhibit T cell immune responses. Some bacteria, such as *Mycobacterium tuberculosis* (*M. tuberculosis*), actively hide from T cells [37], whereas other species, such as *Shigella flexneri (S. flexneri)*, invade T cells and modify their behavior [38]. Other bacteria, such as *Staphylococcus aureus (S. aureus)* secrete exotoxins called superantigens (SAgs), which evolved to target T cells, in order to evoke cellular dysfunction or deletion [39]. SAgs have received much attention in recent years, due to their profound effect on adaptive immunity and their ability to “anergise” large numbers of antigen-specific T cells that were generated by the immune system in the first place, to combat the SAg-producing bacteria [39]. This review examines the latest developments in the past ten years on T cell immunity to bacterial pathogens and also evaluates the mechanisms bacteria use to evade such T cell immunosurveillance.

## 2. How T Cells Fight Bacteria

### 2.1. MHC-Restricted αβ T Cells

To escape the threat of antibody-led defense mechanisms, certain bacteria evolved ways to invade intracellular compartments of mammalian host cells. Bacterial species such as *Listeria monocytogenes (L. monocytogenes)* and *S. flexneri* actively target the cytosol of host cells whilst *M. tuberculosis* and strains of *Salmonella* are known to reside in vacuolar compartments [37]. Other bacteria, such as *S. aureus* and *Streptococcus pyogenes (S. pyogenes)*, which were originally classified as extracellular pathogens, also invade intracellular spaces of phagocytic and non-phagocytic cells in order to escape immune recognition [40,41]. Many intracellular bacteria can prosper inside endothelial cells on mucosal surfaces and in phagocytic APCs, such as dendritic cells (DCs) and macrophages [42]. However, such cellular sub-localization exposes these intracellular pathogens to enzymatic degradation and, consequently, to MHC class I and II antigen processing machinery, which elicit the presentation of bacterial fragments as antigens to αβ CD4^+^ and CD8^+^ T cells. MHC class I-restricted CD8^+^ T cells are critical for clearing bacterial infections and are known to provide protective immunity against a range of bacterial species, including *L. monocytogenes* [43], *M. tuberculosis* [44], *Salmonella enterica serovar typhimurium (S. typhimurium)* [45], and *S. aureu s* [46]. Furthermore, HLA-E restricted CD8^+^ T cells can also engage antigens from *M. tuberculosis* and *S. typhimurium* [21]. MHC class II-restricted CD4^+^ T cells support memory CD8^+^ T cell responses [2,3] and are important for protective immunity against bacterial infections, such as *M. tuberculosis s* [44], *S. aureus* [46], and *S. typhimurium* [47]. In response to infection, naïve CD4^+^ T cells differentiate into distinct “helper” subsets with effector capacity, such as T helper 1 (T_H_1) and T_H_2 cells. These effector CD4^+^ T cell types are distinguished by their roles in cellular or humoral immunity, the cytokines they produce, the transcription factors that control their development, and the types of pathogens they combat. For example, CD4^+^ T_H_1 cells utilise T-bet, STAT1, and STAT4 for transcriptional regulation, produce interferon-γ (IFNγ), tumor necrosis factor-α (TNFα), interleukin-2 (IL-2), and drive cellular immune responses against intracellular pathogens, including bacteria. In contrast, CD4^+^ T_H_2 cells promote humoral immune responses against large extracellular pathogens (e.g., parasites), are transcriptionally controlled by GATA3, STAT5, and STAT6 and produce cytokines such as IL-4, IL-5, and IL-13 [48]. As such, CD4^+^ T_H_ cell subsets provide a vital source of effector cytokines that help CD8^+^ T cells and prime phagocytic cell subsets to kill bacteria. Interestingly, αβ CD4^+^ T cells can also seize and kill bacteria from infected DCs, in a process more akin to innate immune cells [49]. Such “transphagocytic” CD4^+^ T cells can also act as *bona fide* APCs and present bacterial antigens to naïve CD8^+^ T cells and generate memory CD8^+^ T cells [50]. These capabilities position CD4^+^ T cells capable of transcending the boundary between innate and adaptive immunity in a way that is more reminiscent of the unconventional T cell populations. In the following sections, we examine new developments from the past decade in our understanding of how MHC class I restricted CD8^+^ T cells and MHC class II restricted CD4^+^ T cells target bacteria and mediate protective immunity.

#### 2.1.1. CD8^+^ T Cells: The Advent of Tissue Resident Memory T Cell Immunity

Memory CD8^+^ T cells are crucial for generating rapid recall responses to infection and maintaining long-term immunity. Memory CD8^+^ T cell responses were defined at the turn of the century to involve two subsets—central memory T (T_CM_) and effector memory T (T_EM_) cells, based on their tissue homing patterns [51]. CD8^+^ T_CM_ and T_EM_ cells both produce effector cytokines (e.g., IFNγ, TNFα, IL-2) and cytotoxic molecules (e.g., perforin) [52] in response to antigen, although CD8^+^ T_EM_ cells evoke superior protection from bacteria, including *L. monocytogenes* [53,54]. CD8^+^ T_CM_ cells express lymph node homing receptors, notably CCR7 and CD62L, which enable them to track from the blood to secondary lymphoid organs. In contrast, CD8^+^ T_EM_ cells lack CCR7 and preferentially localize in non-lymphoid tissues [55]. However, for many years, it was relatively unknown whether CD8^+^ T_EM_ cells permanently resided in non-lymphoid tissue, after the resolution of infection or recirculated back to blood. Nonetheless, seminal research between 2008 and 2010 changed the memory T cell landscape and determined that memory CD8^+^ T cells could remain in non-lymphoid tissues after virus infection, without the need for replenishment [56,57,58], marking the discovery of tissue resident memory T (T_RM_) cells. Since then, an acceleration of research in humans and mice into CD8^+^ T_RM_ cells (and CD4^+^ T_RM_ cells; see Section 2.1.3) in the past ten years, has greatly improved our knowledge of memory T cell immunity in tissues. It highlighted their critical role in facilitating protective immunity against an array of pathogenic microorganisms [59]. CD8^+^ T_RM_ cells populate mucosal barrier tissues (e.g., lung, intestine, female reproductive tract, salivary glands), which are routinely exposed to infectious pathogens but also inhabit lymph nodes and vital peripheral organs, such as the brain, liver, and kidneys [59]. CD8^+^ T_RM_ cells are phenotypically, transcriptionally, and metabolically distinct from CD8^+^ T_EM_ and T_CM_ cells, and are poised in tissue to mount rapid immune responses against invading pathogens [60,61,62,63,64]. A cardinal feature of CD8^+^ T_RM_ cells is their inability to recirculate through the bloodstream or secondary lymphoid organs. Like CD8^+^ T_EM_ cells, they lack CD62L and CCR7 [62,65], but typically express CD69, the αE integrin CD103 and downregulated levels of sphingosine-1-phosphate receptors and transcription factors controlling migration, which promote their tissue retention [62,66,67]. CD8^+^ T_RM_ cell populations are heterogenous, even within the same tissue. Although CD103 and CD69^−^ T_RM_ cells do exist [61,68,69], they can express other phenotypic markers, such as CXCR6, CD11a, CD49a, CD101, and PD-1 [70,71,72,73], which demarcate these cells in a range of anatomical locations. The development and functionality of CD8^+^ T_RM_ cells does not depend on the presence of antigen in certain tissues [59], but requires the activity of transcription factors, including Hobit [61], Blimp1 [61], Notch1 [71], Runx3 [74], and Bhlhe40 [75]. Additionally, cytokines, such as transforming growth factor-β (TGF-β), IL-15, and IL-33 [20,59,62,67,76,77,78], influence CD8^+^ T_RM_ cell development and help “program” the T_RM_ phenotype. In response to antigen, CD8^+^ T_RM_ cells can locally proliferate, produce effector cytokines (IFNγ, TNFα, IL-2, IL-17) [60,63,70,71], and cytotoxic molecules (granzyme B), can also stimulate the recruitment of circulating memory T cells in recall responses, upon antigen re-encounter [79,80].

CD8^+^ T_RM_ cells provide superior protection against pathogens that invade barrier tissues and mucosal organs, yet the response of CD8^+^ T_RM_ cells to bacterial infection is less defined than that for viral infection. Nevertheless, these cells provide crucial protective immunity against a number of bacterial species. Studies examining the efficacy of vaccines against *M. tuberculosis*, notably the Bacillus Calmette–Guérin vaccine [81,82] and those vectored by replication-incompetent [83] or competent [84] viruses (e.g., cytomegalovirus), demonstrated that lung CD8^+^ T_RM_ cells provide significant protection against tuberculosis, forming a correlate of vaccine-induced immunity. Following infection and re-challenge with *L. monocytogenes*, a food-borne Gram-positive bacterium, intestinal CD69^+^CD103^+^ CD8^+^ T_RM_ cells, respond to mediate local and protective immunity, and rely on α_4_β_7_ integrin to home to the intestine [78]. CD8^+^ T_RM_ cells provide protection against other food-borne bacterium, *Yersinia pseudotuberculosis* (*Y. pseudotuberculosis*), which causes Far East scarlet-like fever. In contrast to CD8^+^ T_RM_ cells that respond to intestinal *L. monocytogenes* infection, protection is not dependent on TGF-β. Indeed, in mouse models of *Y. pseudotuberculosis* infection, intestinal macrophages produce TGF-β-suppressing cytokines (IFN-β, IL-12), which facilitate the development of CD69^+^CD103^−^ CD8^+^ T_RM_ cells that mediate protective immunity [85,86]. These differences highlight specific nuances in CD8^+^ T_RM_ cell driven immunity to bacterial infection, although such protective immunity is not an exclusive property of all bacterial infections as *S. typhimurium*-driven T cell immunity does not depend on the presence of CD8^+^ T_RM_ cells [87,88].

#### 2.1.2. HLA-E Restricted CD8^+^ T Cells: Non-Classical Bacterial Recognition

HLA-E belongs to a family of non-classical MHC class Ib molecules that includes HLA-F, HLA-G, and HFE and possesses two alleles (HLA-E*01:01, HLA-E*01:03) that are encoded in >99% of the global population [21]. HLA-E is found on certain APCs and also in natural killer (NK) cells, where it functions as a ligand for inhibitory CD94/NKG2(A-C) receptors [89] and acts to regulate NK cell activity. Accordingly, HLA-E was thought to be involved in innate immune responses. However, it was determined that HLA-E modulates the activity of CD8^+^ T cells by binding peptide leader sequences [90], and that viruses could mimic [91] such activity. From here, seminal work between 1998 and 2002 provided the first evidence that HLA-E restricted CD8^+^ T cells could respond to *M. tuberculosis* [92,93], heralding a hugely important discovery in the field of non-classical, MHC-restricted T cell recognition of bacteria. It was later found that HLA-E restricted CD8^+^ T cells could target novel bacterial peptide antigens from *S. typhimurium* [94], indicating that HLA-E mediated immunity against bacteria was not restricted to *M. tuberculosis.* In the past decade, a range of approaches including bioinformatics and mass spectrometry discovered a number of HLA-E-specific peptides from *M. tuberculosis* [95,96,97,98,99]. Furthermore, the crystal structure of HLA-E, in complex with an immunogenic peptide from *M. tuberculosis*, was recently solved [100], revealing insights into the structural features of HLA-E-specific antigen presentation. HLA-E is enriched on *M. tuberculosis* phagosomes, enabling the loading of antigenic peptides in HLA-E in infected cells [101] and HLA-E-restricted CD8^+^ T cells mediate recognition, even when *M. tuberculosis* resides in late endosomal vacuoles [102]. Immunological investigations in HLA-E-restricted CD8^+^ T cell responses found that *S. typhimurium*-reactive cells were polyfunctional [103], were comparable in adults and paediatric individuals vaccinated against *S. typhimurium* [104], and correlated with clinical outcome in challenge studies [105]. In patients with *M. tuberculosis* infection, HLA-E restricted CD8^+^ T cells appeared to be more akin to CD4^+^ T_H_2 cells, since they produced T_H_2-specific cytokines (IL-4, IL-5, IL-13) upon antigen stimulation, expressed the transcription factor GATA3, and could activate B cells [95,99]. Furthermore, these cells inhibited *M. tuberculosis* growth, challenging conventional wisdom that “T_H_2-like” cells do not participate in mycobacterial control [99]. These developments in the past decade further indicated that HLA-E drives the protective CD8^+^ T cell responses to bacteria, embodying an important facet of the adaptive immune response to infection.

#### 2.1.3. CD4^+^ T Cells: Anti-Bacterial Immunity Requires “Help”

MHC class II-restricted CD4^+^ T cells are important for protective immunity against a range of bacterial infections and provide a vital source of cytokines that help to facilitate such anti-bacterial immunity. Research on mice deficient in effector cytokines produced by CD4^+^ T_H_ cell subsets [48] or humans with cytokine receptor deficiencies [106] illustrated the fundamental role that these cytokines and specific CD4^+^ T cell types play in this process. CD4^+^ T_H_1 and T_H_2 cells are the most categorized subsets in CD4^+^ T cell biology and were thought to be the only types that existed until the identification of CD4^+^ regulatory T cells in the early 2000s, and CD4^+^ T_H_17 cells during 2005 and 2006 [4]. Unlike CD4^+^ T_H_1 or T_H_2 cells, CD4^+^ T_H_17 cells express RORγt and STAT3 transcription factors, which control their lineage commitment and secrete IL-17A, IL-17F, and IL-22 as effector cytokines [48]. Since their identification, it has become well-known that CD4^+^ T_H_17 cells are fundamentally important for driving immune responses against bacterial infection, along with CD4^+^ T_H_1 subsets. As such, IFNγ and TNFα producing CD4^+^ T_H_1 cells provide protective immunity against *Francisella tularensis (F. tularensis)* [107], whilst CD4^+^ T_H_17 cells contribute to immune defense mechanisms directed against *Mycobacterium bovis* (*M. bovis*) [108] and *Streptococcus pneumoniae* (*S. pneumoniae*) [109]. However, both CD4^+^ T_H_1 and T_H_17 cells elicit immune responses against *M. tuberculosis* [110,111], *L. monocytogenes* [112,113], *Bordetella pertussis* (*B. pertussis*) [114], and *S. aureus* [115,116,117] infections. Such anti-bacterial immunity driven by CD4^+^ T cell appears to be less dependent on CD4^+^ T_H_2 cells, which drive humoral responses against pathogens, such as helminths and parasites. However, since 2009, it was established that a distinct subset of CD4^+^ T cells, called T follicular helper (T_FH_) cells [118,119,120], helped to support humoral immunity against pathogens that invade intracellular compartments, such as viruses and bacteria. Since then, extensive research into CD4^+^ T_FH_ cells in humans and mice in the past decade has evolved our knowledge of these cells and the mechanisms of adaptive immunity that they drive. CD4^+^ T_FH_ cells support antibody production in germinal centers (GCs) of secondary lymphoid organs (e.g., lymph nodes, tonsils), by promoting immunoglobulin class switching, B cell affinity maturation, and memory B cell responses [4,121]. CD4^+^ T_FH_ cells can also migrate to tertiary lymphoid structures (TLS) in non-lymphoid organs (e.g., lung) and possess a phenotypic, transcriptomic, and functional profile that separates them from other CD4^+^ Th cell subsets. CD4^+^ T_FH_ cells express PD-1, ICOS, and high levels of CXCR5 [122], which enables their migration to B-cell-rich follicles of lymphoid organs [123]. GC-resident CD4^+^ T_FH_ cells depend on Bcl-6 to drive their differentiation [118,119,120,121], which acts to block commitment to non-T_FH_ cell lineages, by repressing transcription factors that govern T_H_1, T_H_2, and T_H_17 cell fate, such as T-bet and GATA3, and genes that are important for their effector function, such as *Ifng* and *Il17a* [118,119,120,124,125]. CD4^+^ T_FH_ cells produce IL-21 and IL-4 [4,121] as effector cytokines and their development and function is influenced by TCR stimulation and other cytokines, such as IL-12 [126] and IL-27 [127]. CD4^+^ T_FH_-like cell populations exist in the blood [128], which are clonotypically similar to GC-resident CD4^+^ T_FH_ cells [129] although these cells lack Bcl-6 [121] and display lower levels of PD-1, CXCR5, and ICOS [121]. CD4^+^ T_FH_ cells play a vital protective role against viral infection, yet there is evidence that these cells support protective immune responses against bacteria. CD4^+^ T_FH_ cells promote humoral immune responses against intestinal *Citrobacter rodentium* infection in mice [130] and mediate protective immunity against *M. tuberculosis* infection, by accumulating within TLS in lung granulomas in humans, mice, and non-human primates [131]. However, this role for CD4^+^ T_FH_ cells in anti-bacterial immunity is not fully established and might be specific to distinct bacterial species. GC-resident CD4^+^ T_FH_ cells in children with recurrent tonsillitis, caused by *S. pyogenes* infection, can become phenotypically skewed into pathogenic B cell killing effectors, which reduces humoral immunity as a consequence [132]. Thus, further research into the anti-bacterial role for CD4^+^ T_FH_ cells is of importance.

Much like memory CD8^+^ T cells, CD4^+^ T cells form long-lived circulating (T_CM_, T_EM_) and tissue-resident (T_RM_) memory populations, which establish upon the resolution of infection. CD4^+^ T_RM_ cells share many cardinal features of CD8^+^ T_RM_ cells—they populate barrier sites and mucosal organs (e.g., lung, skin, salivary glands, intestine), they mediate local protection against infection, they phenotypically express CD69 and CD11a [133] and share a core transcriptional signature [134]. However, CD4^+^ T_RM_ cells lack, or possess low levels of, CD103 and can depend on receiving different signals (e.g., antigen), even in the same anatomical site [135,136,137,138]. Research in the past decade determined that CD4^+^ T_RM_ cells are induced at various mucosal sites to mediate protective immunity against an array of bacterial pathogens, including *M. tuberculosis* [139], *B. pertussis* [140], *S. pneumoniae* [141], *L. monocytogenes* [142], *C. rodentium* [143], and *Chlamydia trachomatis* [144]. Despite this, less is known about CD4^+^ T_RM_ cells in infection settings, in comparison to their CD8^+^ T_RM_ cell counterparts, or whether CD4^+^ T_RM_ cells form effector T_H_ cell subtypes, such as T_H_1, T_H_2, and T_H_17, or even T_FH_ cells. Recent evidence provided some insight, indicating that T_H_1 [145] and T_H_17-like [146] CD4^+^ T_RM_ cells can mobilize in response to infection. However, the factors governing the development of these effector CD4^+^ T_RM_ cell subtypes, especially in the context of memory responses to pathogen re-exposure, requires more definition.

### 2.2. Unconventional αβ and γδ T Cells

#### 2.2.1. CD1-Restricted T Cells: Lipid-Driven T Cell Immunosurveillance

CD1 molecules are non-polymorphic, antigen-presenting molecules encoded outside of the MHC locus that have evolved to present microbial lipids to T cells. In humans, there are 4 CD1 protein isoforms (CD1a-d), which are capable of presenting antigen that are apportioned into two groups [147]: CD1a, CD1b, and CD1c form group 1, CD1d is the sole member of group 2, as its expression is constitutive rather than inducible [23,24]. These isoforms display distinct patterns of expression on a range of APCs, although all are found on myeloid DCs, and traffic differentially through endosomes and lysosomes to acquire antigens, each with highly hydrophobic antigen binding grooves that are capable of presenting lipids [23,24]. Antigen-specific, CD1-restricted T cells greatly outnumber αβ T cells reactive to a specific pMHC complex [24]. Seminal research between 1989 and 1994 discovered that αβ and γδ T cells were activated by CD1 molecules [148] and that lipid antigens from *M. tuberculosis* elicited CD1b-restricted T cell responses [22,149]. Many more CD1-restricted lipid antigens of microbial origin were since determined and the identity of T cells that react to them has become clearer. The development of CD1d-specific tetramers in the early 2000s determined that NKT cells could respond to CD1d-presented lipid antigens. Two classes of NKT cells exist—type I iNKT cells and type II “diverse” NKT cells. Type I iNKT cells possess TCR α-chains encoding fixed TRAV–TCR α-chain J (TRAJ) rearrangements (TRAV18/TRAJ10 in humans), which pair with TCR β-chains displaying limited TRBV gene usage, whilst type II NKT cells possess more polyclonal TCR repertoires composed of cells expressing αβ, γδ, or δ/αβ TCRs [24,33,150,151,152,153]. Microbial lipid antigens that are structurally similar to α-galactosylceramide, a glycolipid from the marine sponge *Agelas mauritianus* [154], are potent ligands for type I iNKT cells [24,153], but not type II NKT cells [150,151]. Type I iNKT cells respond rapidly to antigen and exist in diverse subsets defined by their effector cytokine production and transcription factor profile [24]. Type I iNKT cells play a key role in driving antigen-specific immunity against bacterial pathogens, such as *M. bovis* [155], *Sphingomonas capsulate* [156], *Borrelia burgdorferi (B. burgdorferi)* [157], and *S. pneumoniae* [158]. However, the contribution of type II NKT cells in anti-bacterial immunity is less defined, although these cells are activated by phospholipid antigens from *M. tuberculosis* and *Corynebacterium glutamicum* [159].

Up until the past decade, research into group 1 CD1-restricted T cells was slow, hampered by a lack of reagents that could detect CD1a-, CD1b-, and CD1c-reactive T cells and because mice do not encode group 1 CD1 molecules. However, the advent of tetramers specific to CD1a [160], CD1b [161], and CD1c [162] in the past ten years, changed the landscape and identified group 1 CD1-restricted T cells that are responsive to lipid antigens from bacteria. Two types of CD1b-restricted T cells; those bearing TRBV4-1^+^ αβ TCRs [161] or those encoding invariant TCR α-chains (“germline-encoded mycolyl lipid-reactive” T cells) [163], were found to be responsive to glucose-6-O-monomycolate from *M. tuberculosis*. CD1c-restricted T cells exhibiting TRBV7-8 or TRBV7-9^+^ αβ TCRs engage phosphomycoketide antigens from *M. tuberculosis* [164] and such CD1c-specific antigen reactivity was not exclusive to αβ TCRs, as bacterial lipid reactive, CD1c-restricted γδ TCRs exist [32]. Studies in human group 1 CD1 transgenic (hCD1tg) mice demonstrated that CD1b-restricted T cells contribute to protective immunity against *M. tuberculosis* [165], and high numbers of CD1b-restricted T cells are found in humans with active TB infection [166]. CD1-presented lipid antigens were also discovered in other bacterial species, such as *B. burgdorferi* [167], *Brucella melitensis, Salmonella typhimurium*, and *S. aureus* [168]. Recently, an in vivo model of systemic *S. aureus* infection in hCD1tg mice determined that group 1 CD1-restricted T cells elicit protective immunity, by responding to lipid antigens from *S. aureus* [169]. As such, these recent developments highlight an integral role for CD1-restricted T cells or NKT cells in anti-bacterial immunity and the key involvement of CD1 molecules is in presenting lipidic antigens to these specialised T cell or NKT cell subsets.

#### 2.2.2. γδ T Cells: A Story of Anti-Bacterial Immunity Bound by Two Divergent TCR δ-Chains

γδ T cells are an unconventional T cell population that represent a small fraction (0.5–10%) of circulating CD3^+^ T cells in adult humans, and display immunological features common to both innate and adaptive immune systems, rendering them to be often referred to as innate-like lymphocytes [24]. γδ T cells express TCRs composed of rearranged TCR γ- and TCR δ-chains that are distinct from TCRs found on αβ T cells, and are encoded by lower numbers of V, D, and J segments, which limit their clonotypic diversity. Indeed, the restricted repertoire of TCR γ-chain V (TRGV) and TCR δ-chain V (TRDV) gene segments available for rearrangement (e.g., only 9 functional TRGV genes) led to theories that γδ TCRs identify conserved self-proteins of low variability [170]. During embryonic development, γδ T cells formed the first T cell population that was created and these cells established in low numbers in the circulation, but infiltrated peripheral mucosal and epithelial tissues at higher frequency. γδ T cells rapidly produced effector cytokines (e.g., IFNγ, TNFα, IL-17) in response to differing pathologies, including bacterial infection [171], where such cytokine production helped to recruit neutrophils, enhance adaptive immune defenses, and provided protection from bacterial invasion. There is evidence that γδ T cells form long-lived memory populations upon bacterial infection, some of which permanently populate the affected tissues, after pathogen clearance. In vivo studies in rodents and primates showed that γδ T cells provide protective memory responses against a number of bacterial pathogens, such as *M. tuberculosis*, *S. aureus*, *B. pertussis*, *L. monocytogenes*, and *Salmonella enterica* (reviewed in [171]). Such immunity was driven by clonotypically distinct γδ T cell subsets possessing effector functionality in different anatomical locations, such as the blood, lungs, peritoneum, skin, and intestine. γδ T cells are often influenced by the microbiota found at barrier sites and the regulation of γδ T cells by commensal bacteria can help drive protective immune responses against pathogenic bacteria, such as *Pseudomonas aeruginosa* [172].

In humans, γδ T cells can be classified into two main populations, based on their expression of TCR δ-chains encoded by two TRDV genes—TRDV1^+^ γδ T cells (or Vδ1^+^ T cells) and TRDV2^+^ γδ T cells (or Vδ2^+^ T cells). Vδ1^+^ T cells are abundant in the skin, intestine, and uterus, whereas Vδ2^+^ T cells constitute the majority of peripheral blood γδ T cells [24,173,174]. Vδ2^+^ T cells are dominated by TCR γδ chain rearrangements encoded by the TRGV9 and TRDV2 segments, which almost exclusively pair together, forming the more frequently termed “Vγ9^+^Vδ2^+”^ T cells that make up large proportions of peripheral blood γδ T cells in humans and other species [175]. Vγ9^+^Vδ2^+^ T cells are enriched in fetal peripheral blood, as “pre-programmed”, semi-invariant effector T cells [176] with innate-like capacity, and these cells are known to be responsive to prenyl pyrophosphate metabolites (or “phosphoantigens” (PAgs)), from the eukaryotic mevalonate pathway [177] that are produced by mammalian cells. However, PAgs are also produced by the non-mevalonate pathway in bacterial pathogens, such as *M. tuberculosis* [178] and *Escherichia coli (E. coli*) [179], endowing Vγ9^+^Vδ2^+^ T cells with the ability to ‘sense’ infected cells [175]. For a while, the mode of PAg presentation to Vγ9^+^Vδ2^+^ T cells remained unsolved, yet recent work in the past decade determined that PAgs activate Vγ9^+^Vδ2^+^ T cells by binding to BTN3A1 [180,181,182] and that BTN3A2 [183] is an important factor for regulating such activation. The molecular mechanisms governing Vγ9^+^Vδ2^+^ TCR recognition of PAgs remained unclear until very recently when it was discovered that BTN2A1 acts a ligand for Vγ9^+^Vδ2^+^ T cells, acting to bind germline-encoded regions of TRGV9 and working in concert with BTN3A1, to promote Vγ9^+^Vδ2^+^ T cell responses to PAgs [184,185]. Unlike Vγ9^+^Vδ2^+^ T cells, peripheral blood Vγ9^-^Vδ2^+^ T cells are unresponsive to PAgs, existing as a divergent, adaptive-like subset of Vδ2^+^ T cells that is found at much lower frequency, but undergoes clonal expansion and effector differentiation in response to viral infection [174]. In humans, Vδ1^+^ T cells also constitute a minority of the adult peripheral blood γδ T cell population [186], acting more to populate epithelial and mucosal tissues, particularly the intestine [187]. Vδ1^+^ T cells evolved a distinct biology from Vδ2^+^ T cell subsets and are shaped by TCR-dependent clonal selection [186]. Murine Vδ1^+^ T cells display broader TRGV usage patterns and utilize gene segments that are distinctive to individual tissue sites (e.g., TRDV5^+^ Vδ1^+^ T cells in the skin, TRGV7^+^ or TRGV1^+^ δ1^+^ T cells in the intestine) [24]. Recent research showed that human intestinal Vδ1^+^ T cell subsets utilize TCR γ-chains encoding TRGV4 and are regulated by BTNL3 and BTNL8 molecules on the surface of gut epithelia [188]. BTNL3 was shown to bind germline-encoded regions of TRGV4 encoding TCRs [189], providing a mode of antigen recognition for Vδ1^+^ TCRs. However, Vδ1^+^ T cells appear to be promiscuous in their antigen-binding capabilities, possessing the ability to engage with pMHC class I complexes [35], and MHC-like molecules or homologs, such as CD1c [32,148], CD1d [30,31,33], and MICA [34]. Indeed, recently it was found that clonally diverse Vδ1^+^ TCRs could bind MR1 underneath the antigen-binding cleft [29], redefining our understanding of TCR recognition and identifying MR1 as a self-ligand for Vδ1^+^ T cells. Such MR1-restricted γδ T cells were not found in Vγ9^+^Vδ2^+^ T cell populations and populated damaged tissues, suggesting that they possess roles in pathology. Further studies are required to define how MR1-restricted γδ T cells facilitate immune responses in pathological settings, but also to learn more about the molecular determinants of antigen recognition, notably of microbial ligands, through Vδ1^+^ T cells.

#### 2.2.3. MAIT Cells: Masters of Riboflavin-Driven T Cell Immunity

MAIT cells are an innate-like, unconventional T cells that respond to MR1-restricted bacterial metabolite antigens and provide a powerful source of pro-inflammatory cytokines, relative to their frequency in T cell populations [24,36,190]. In 1993, MAIT cells were first identified as a CD4^-^CD8^-^ (DN) T cell population, expressing an invariant, TRAV1-2/TRAJ33 encoded a TCR α-chain [191] that was later found to pair with a limited repertoire of TCR β-chains (TRBV20-1 or TRBV6 family genes in humans) that are conserved across mammals [192]. In 2003, a landmark study that devised the term “MAIT cells”, demonstrated the specificity of MAIT cells to MR1 and their enrichment in mucosal locations, such as the gut lamina propria [193]. For many years after, the function of MAIT cells and the antigens they respond to, remained unknown. However, the development of monoclonal antibodies that could detect TRAV1-2 [194] or block MR1-facilitated seminal discoveries into the function of MAIT cells, which in 2010 was proved to be capable of responding to a range of Gram-negative and Gram-positive bacterial species, including *Klebsiella pneumoniae* (*K. pneumoniae*)*, E. coli*, and *S. aureus* [195,196]. Such reactivity was not exclusive to all bacteria, which indicated that such microbial sensing was potentially driven by recognition of common antigen(s). Since then, the past decade witnessed an exceptional surge in research centered around understanding the role and function of MAIT cells in combating bacterial infections and also on the process through which these cells develop in the periphery, the phenotypic and transcriptomic signatures they possess, the antigens they engage, and the activation profiles they generate. In 2012, an important breakthrough was made in the identity of MAIT cell reactive antigens, which determined that MR1 could present metabolites of vitamin B2 (riboflavin) and B9 (folic acid) from bacterial species to MAIT cells [25]. This significant discovery paved way for the identification of key ribityllumazine- and pyrimidine-based metabolite antigens of the microbial riboflavin biosynthesis pathway, including 5-OP-RU (5-(2-oxopropylideneamino)-6-D-ribitylaminouracil), which could potently activate MAIT cells [25,197]. Bacterial species that activate MAIT cells possess intact riboflavin biosynthesis, and such dependency on antigens from this pathway was verified in experiments where deletion of key pathway enzymes rendered bacteria unresponsive to MAIT cells [25,190,197]. Furthermore, many structural-based studies examined the precise mechanism involved in the recognition of MR1-bound ribityllumazine- and pyrimidine-based antigens through MAIT-specific TCRs [190] and suggested that these metabolite antigens did not fill the MR1 antigen-binding cleft, which opened up the possibility that other antigens could be presented by MR1 to MAIT cells. Indeed, further research determined that MR1 could present non-riboflavin-derived microbial antigens [198,199] and possessed enough flexibility to accommodate the binding of anti-inflammatory drugs [200]. Nonetheless, such identification of bacterial metabolite antigens led to the development of human and mouse MR1 tetramers [201,202], loaded with antigens such as 5-OP-RU. Such a technological advancement helped revolutionize our understanding of these cells, how they develop in the thymus, and also the bacterial infections they are mobilized to fight against.

In humans, MAIT cells are now typically defined as 5-OP-RU/MR1 tetramer binding T cells that bear a TRAV1-2/TRAJ33 encoded TCR α-chain [190], although “non-classical” MAIT cells that possess TRAV1-2^-^ TCR α-chains but are found at lower frequency, do exist. 5-OP-RU/MR1 tetramer^+^ MAIT cells also encode TCR α-chains with alternative TRAJ genes (e.g., TRAJ20, 12) in humans [202] and mice, notably those deficient in TRAJ33 [203]. The majority of MAIT cells represent CD8αα, CD8αβ^+^, or DN T cell subsets, which make up 1–10% of all blood T cells and up to 50% of T cells found in mucosal organs [190]. MAIT cells display an effector memory (CD45RO^+^CCR7^-^CD62L^-^) phenotype and typically express high levels of CD161, found to be more than a 100-fold higher than that expressed by many other T cell types. MAIT cells express many chemokine receptors, including CCR5 and CCR6, which enable them to recirculate between the circulation and tissues [204], and upon TCR-mediated recognition of antigen, produce effector cytokines (IFNγ, TNFα, IL-2, IL-17, IL-22) as part of defined effector subsets (e.g., MAIT1 cells produce IFNγ, MAIT17 cells produce IL-17), and gain cytotoxic capabilities [190]. MAIT cells express transcription factors, such as PZLF1, T-bet, Eomes, RORγt, and Helios, which drive functionality (e.g., T-bet and MAIT1 cells, RORγt and MAIT17 cells) and share many phenotypic similarities with NK cells [190]. Many studies showed that in the absence of MR1-presented antigen, MAIT cells can be activated by cytokines, such as IL-12 and IL-18 [205], whose receptors are found at high levels on the cell surface and produce efficient activation in synergy, rather than alone. Such TCR-independent activation is observed during bacterial [190] and also viral [206] infection, broadening the role that MAIT cells play in immune responses to bacteria carrying intact riboflavin biosynthesis to infectious situations where IL-12 or IL-18 is produced [205]. More recently, it was shown that IL-23 co-stimulates antigen-specific MAIT cells during pulmonary bacterial infection and drives their proliferation and differentiation [207]. Interestingly, CD80/86-CD28 signaling was found to be not instrumental for MAIT cell expansion, in contravention of the rules for conventional T cells and iNKT cells [207]. A prominent feature of MAIT cells is their capability to exhibit effector functions, prior to exiting the thymus. The use of 5-OP-RU/MR1 tetramers helped to determine that MAIT cells follow a three-stage intrathymic developmental pathway in humans and mice [208]. A recent study determined that commensal bacteria are vital for intrathymic development of MAIT cells in mice, providing a source of riboflavin metabolites that are transferred to the thymus to help direct MAIT cell production [209]. Such synergy between MAIT cells and riboflavin-producing commensals, during early life, depends on a small temporal window that functions to imprint MAIT cell abundance in the periphery [210]. In response to bacterial pathogens, MAIT cells provide a fundamental cellular population that is designed to limit infection. Studies using human blood samples, *Mr1^-/-^* mice and other mouse models (e.g., MAIT TCR transgenic mice) generated for studying MAIT cell control of bacterial infection in vivo, revealed the true landscape of infections that MAIT cells mobilize against, to provide protective immunity. MAIT cells were found to be induced or activated in response to infection from a large repertoire of bacterial species that included *Clostridium difficile*, *E. coli*, *F. tularensis*, *Haemophilius influenzae*, *K. pneumoniae*, L*egionella longbeachae*, *Mycobacterium abscessus*, *M. tuberculosis*, *S. flexneri*, *S. typhimurium*, and *S. pneumoniae* (reviewed in [190]). Whilst this protective role was highly apparent, MAIT cells were showed to also have a pathogenic role, notably during *Helicobacter pylori* infection [211]. As such, it is important to learn more about the protective or pathogenic roles of MAIT cells in response to bacterial infection.

## 3. Bacterial Immune Evasion Strategies

### 3.1. A Pathogenic Game of ‘Hide and Seek’ and T Cell Evasion by M. tuberculosis

A key characteristic of many pathogens is persistence—the continued presence of pathogen in environments that are considered stressful or hostile conditions, which might have limited nutrients and might be shared with antimicrobial regents or threatening immune cells. During persistence, the pathogen is non-infectious, having stopped progressive activities, such as cell development and reproduction, and thus remains undetected by the host, while it ‘hides’ in a non-replicating state [212]. Until a more comfortable environment can be secured, such persistence will continue where the pathogen remains viable but does not thrive. However, the pathogen can play ‘hide and seek’ and re-appear once the immune system is evaded and possibly deceived at the infection or colonization site. One pathogenic bacterium that excels at this is *M. tuberculosis* [213], which evades host immune responses in order to establish a chronic infection. *M. tuberculosis* is a facultative intracellular pathogen that is known to reside inside a variety of APCs, including macrophage and DC subsets. Following inhalation of *M. tuberculosis* droplets, the bacteria specifically targets alveolar macrophages that do not robustly respond or detect the infection, resulting in a dampened innate immune response and delayed activation of adaptive immunity [214]. Indeed, murine models established that the triggering of antigen-specific T cells following aerosol infection with *M. tuberculosis* is delayed, relative to that of other pathogens such as the influenza virus [215]. As such, the delay in immune triggering that is manufactured by *M. tuberculosis* is unique amongst lung bacterial pathogens, suggesting that *M. tuberculosis* actively uses the alveolar macrophages to avoid rapid detection. Upon invading host cells through phagocytosis, *M. tuberculosis* can replicate within the infected cells by arresting phagosome maturation [216]. This is accomplished by *M. tuberculosis* changing its composition, as the structure of the cell wall and specific molecules on its surface serve as a barrier that allows the macrophages to maintain a neutral pH [217]. This mechanism allows the pathogen to avoid exposure to lysosomal hydrolases, unfavorable low pH conditions produced by the immune response, and other bactericidal lysosomal components [212]. Additionally, *M. tuberculosis* is capable of producing factors that modulate the expression of pro-apoptotic and anti-apoptotic genes in macrophages [218], which has implications for innate immune responses. Inhibition of apoptosis might be a major mechanism, whereby *M. tuberculosis* delays the acquisition of bacteria by DCs and the onset of adaptive immunity. It was also established that *M. tuberculosis*-infected macrophages preferentially synthesize lipoxin A4 [219,220], which helps create an anti-apoptotic environment that delays the onset of CD4^+^ and CD8^+^ T cell responses.

*M. tuberculosis* not only hides from the immune system but can also modulate adaptive immune responses by inhibiting T cell activities. *M. tuberculosis* can chronically stimulate antigen-specific CD4^+^ T cells (i.e., ESAT6-specific) to drive functional exhaustion [221], whilst equally reducing the expression of antigens targeted by other CD4^+^ T cell populations to evade detection [222,223]. *M. tuberculosis* can suppress T cell responses by recruiting mesenchymal stem cells to the site of infection, which produce nitric oxide to inhibit T cell activity [224]. *M. tuberculosis* possesses antigens that are recognised by FoxP3^+^CD4^+^ regulatory T cells, which expand in patients with an active infection and act to suppress the effector T cell responses against the bacterium, enabling the infection to expand and prolonging the bacterial burden [225,226]. *M. tuberculosis* does not evolve rapidly and, as a result, the antigens that drive *M. tuberculosis*-specific T cell responses are conserved across the human population [227]—a striking contrast to influenza or HIV that actively evades T cell immunity by mutating antigens to avoid detection [212]. It is also thought that defective interactions between T cells and cells harbouring *M. tuberculosis* contribute to the failure of T cells to remove infection. The ability to image *M. tuberculosis*-induced T cells and their real-time interactions with infected cells in tissues is likely to guide the design and selection of TB vaccines [228]. No global approaches were previously employed to systemically define the repertoire of *M. tuberculosis* immune evasion genes, especially those that enable escape from T cell surveillance. The distinct mechanisms used by *M. tuberculosis* to avoid detection, and thus, extinction in a host, can now be explored in great detail, using global approaches to modify host genetics or to integrate diversity into host models. In particular, the development of CRISPR-Cas gene editing mechanisms that generate “loss of function” and “gain of function” systems, might be particularly useful in understanding the mechanisms used by *M. tuberculosis* for immune evasion [229] and is a likely avenue of future research.

### 3.2. Type III Secretion Systems and Pore-Forming Toxins: Secretion-Driven Bacterial Immune Defense

Pathogenic bacteria evolved a wide repertoire of virulence mechanisms that promote immune evasion and bacterial persistence, including the use of type III secretion systems (T3SSs). T3SSs are complex, macromolecular transport machines found on many pathogenic Gram-negative bacteria [230], including members of *Yersinia, Shigella*, and *Salmonella* species, which subvert host cell immune responses by injecting bacterial effector proteins directly into the host cell, in order to modify their functionality [231]. It was originally thought that the species of *Shigella* and *Salmonella* used T3SSs to gain entry to cells. However, research evolved to show that T3SSs work at a different level, by altering the phagocytic properties of macrophages and possibly their killing capacities [232]. As such, proteins secreted by T3SSs can be classed as manipulators of innate defense mechanisms [233], endowing bacterial pathogens with the ability to alter inflammatory responses from within phagocytic cells. *Shigella* use T3SSs to secrete effector proteins, such as IpaB [234], which binds to and activates Caspase-1 [235] in macrophages, through a process involving the IPAF/ASC inflammasome, which enables them to evade the phagosome and induce pyroptosis [236]. Such exploitation of the IPAF/ASC inflammasome is thought to help *Shigella* to escape macrophages, enabling them to invade the intestinal epithelium [237]. The evasive techniques employed here by *Shigella*, enable it to establish infectious processes [238]. Moreover, *Shigella* evolved to inhibit the production of certain antimicrobial peptides, which are key effector molecules in bacterial host defense. Indeed, early in *Shigella* infections, expression of peptides LL-37 and human β-defensin-1 were found to be dramatically reduced or turned off [239]. This downregulation of immediate defense effectors might encourage bacterial adherence and invasion into host epithelium, and could be an important virulence parameter. Such T3SS-driven evasion strategies act to guard against innate immunity, but in terms of defending against adaptive immune responses, bacteria can use T3SSs to modify T cell behavior. *S. flexneri* evolved to use T3SSs to invade CD4^+^ T cells in order to “paralyze” their migratory patterns and utilizes injected effector proteins to induce inhibitory signals that alter cellular dynamics [38]. *Salmonella enterica* serovars use a unique T3SS that is capable of injecting up to 30 effector proteins with the ability to disrupt cell signaling pathways, interfere with MHC-dependent antigen presentation in DCs [240], and slow the migration of infected DCs [241] with a downstream effect on T cell activation. *Salmonella*-encoded T3SSs can also directly contact T cells, inhibit their proliferation [242] and augment co-inhibitory signaling (e.g., PD-1/PD-L1) between T cells and APCs [243]. *S. aureus*, which displays various levels of virulence, can manipulate host T cell responses that limit bacterial growth but do not eliminate the pathogen during persistent infections. Along with producing potent, T-cell-targeting SAgs (reviewed in Section 3.3), *S. aureus* produces extracellular, pore-forming toxins that lyse T cells upon cellular engagement. *S. aureus* α-toxin forms heptameric pores that destroy T cells [244], whilst leukocidin DE binds to CCR5 to kill T cells [245]. Furthermore, during persistent *S. aureus* infection, T cells can become anergic through a failure in TCR signaling events, which render the T cells unable to respond to antigenic stimulation [246]. The presence of T3SSs and pore-forming toxins, arms bacterial pathogens, such as *Shigella* and *S. aureus*, with an attack and evasion “arsenal”, which acts to dampen host innate and adaptive immune responses and aids their virulence and survival.

### 3.3. SAgs: Microbial Super-Agents That Manipulate T Cell Immunity

SAgs are potent enterotoxins secreted by distinct Gram-positive bacteria, including *S. aureus* and *S. pyogenes*, which target T cells, evoking mechanisms of cellular dysfunction and the initiation of a cytokine storm (involving large amounts of IFNγ, TNFα, and IL-2), which can activate up to 20% of the T cell population [39]. In the late 1960s, the first SAg was identified from the secreted toxins of *S. aureus* and was named staphylococcal enterotoxin A (SEA) for its strong enterotoxic properties. Staphylococcal enterotoxins (SEs) are the causative agents in staphylococcal food poisoning and induce vomiting and diarrhea shortly after ingestion; an effect which is normally self-limiting. However, it is now known that SAgs from both *S. aureus* and *S. pyogenes* are strongly associated with life-threatening conditions, such as pneumonia, sepsis, and toxic shock syndrome (TSS) [39]. Following their initial identification, the mitogenic activity of SEs was not uncovered for some time, with the term ‘superantigen’ not being coined until 1989 when it was discovered that the profound effect of SAgs on the immune system was a result of an immense expansion of T cells bearing the same TCR Vβ domain [247]. This suggested that the interaction between SAgs and TCR Vβ domains was not completely indiscriminate, as only specific TCR Vβ domains could be engaged, leaving only a minority of T cells susceptible to triggering by any given SAg [248]. However, such an act exposes “holes” in the repertoire of T cells available to protect against infection [247], which helps to enhance bacterial colonization [249], providing a bonafide mechanism for bacterial persistence and immune evasion. Since then, many research studies used a range of techniques, including x-ray crystallography, to confirm that SAgs are able to bind and cross-link human αβ TCRs on CD4^+^ or CD8^+^ T cells to HLA class II molecules on APCs, through a variety of structural mechanisms [247,250,251,252,253,254] that bypass the laws of conventional TCR-pMHC interactions. Such a physical interaction between TCRs and SAgs, which exists outside of the peptide-binding groove, involves a large number of TRBV region genes that are targeted in distinct patterns by bacterial SAgs [39,255]. At present, 26 different SAgs from *S. aureus* are described, comprising toxic shock syndrome toxin-1 (TSST-1), 11 SEs (SEA–SEE, SEG–SEI, SER–SET), and 14 SE-like (SEl) proteins (SElJ–SElQ, SElU–SElZ) [256,257]. There are also 14 Streptococcal pyrogenic exotoxins (Spe), which include SpeA, SpeC, SpeG, SpeH, streptococcal mitogenic exotoxin Z, and streptococcal superantigen A [258]. Such TCR Vβ region-SAg interactions are indicated in Figure 2, highlighting the known relationships between specific TRBV genes and a number (but not all) of SAgs from *S.* aureus and *S. pyogenes*. To date, all SAgs appear to specifically target TRBV genes, with the exception of SEH, which interacts with TRAV27^+^ TCR α-chains, whilst also mediating contact with TRBV19 [253,254,259]. It is plausible that SEH might also bind other TRAV genes, however, this is yet to be confirmed.

The ability for SAgs to drive cellular dysfunction or deletion in T cell populations is highly affected by HLA class II polymorphisms [260], which influence patient outcome in infectious diseases driven by these toxins [261]. Some SAgs display a stronger binding affinity to specific HLA class II alleles (e.g., SpeA and HLA-DQB1*06:02) and susceptibility to TSS and invasive forms of group A streptococcal (GAS) infection (e.g., *S. pyogenes*) appears to be correlative with HLA-DQB1*06:02 expression [132]. The physical binding relationship between SAgs, TCRs, and HLA class II alleles was universally accepted for over two decades, until the discovery that SAgs were capable of binding T cell co-stimulatory molecules, notably CD28 [262], and their ligands (i.e., B7-2 [263]) on APCs. Prior to this, co-stimulatory pathways were not thought to bind pathogens or virulence factors [264], and such engagement potently enhanced the avidity between B7-2 and CD28, thereby inducing T-cell hyperactivation, which is a cardinal feature of SAg activity [263]. These binding features position superantigens as biological molecules that are capable of forming quaternary complexes between T cells and APCs, providing evidence of further complexity that was originally unknown. Thus, it provides the necessary scope for further research that examines the structural mechanics behind how these biological toxins interact with T cells.

SAg-mediated illnesses, such as TSS, are associated with immunosuppression upon bacterial invasion, threatening the ability of the host to fight the invading pathogens. It is well-established that the introduction of SAgs leads to an initial expansion of SAg-reactive T cells, yet following this, a large percentage of these T cells are eliminated by apoptosis or become anergic [265]. Such events appear to occur more in memory T cells, consequently reducing the available pool of antigen-experienced T cells [266,267], which creates a state of immune suppression that enables opportunistic infections to thrive. Staphylococcal SAgs are a causative agent of severe pneumonia, acting to induce damage to the pulmonary epithelium and to disrupt mechanisms of neutrophil activation that results in severe immunopathology [268]. The ability of *S. aureus* to form biofilms, combined with the emergence of multidrug-resistant strains and its ability to produce Sags, amplifies the challenge faced when treating infections. As such, the frequency of SAg-producing infections is increasing. *S. aureus* and GAS infections routinely cause recurrent tonsillitis (RT) or tonsillar hyperplasia and drive TCR repertoire skewing patterns that are associated to SAg activity [269]. Children with RT caused by GAS infections display lower titres of antibodies against SpeA, compared to those with singular bouts of tonsillitis, and possess GC-resident CD4^+^ T_FH_ cells that have become phenotypically skewed into pathogenic B cell killing effectors in the tonsil [132]. Studies showed that SAgs can function to subvert T cell immunity to viruses (e.g., influenza) [270,271], which has profound implications for patients with viral/bacterial co-infections. However, recent evidence opposing this view has emerged, suggesting that SAgs (i.e., SEB) could expand influenza-specific CD8^+^ T cells when administered in vivo in mice, prior to virus infection, can re-organize the hierarchical pattern of primary CD8^+^ T cell responses, and promote improved recall immunity [272]. Such an effect was dependent on SAg pre-exposure, as a contrasting phenotype was seen when SEB was administered following a pre-existing influenza infection, highlighting the potential core differences in how SAgs can function when exposed to naïve or memory virus-specific CD8^+^ T cells. SAgs possess the ability to bind αβ TCRs on MHC-restricted CD4^+^ or CD8^+^ T cells, but also target unconventional, antimicrobial T cell populations bearing αβ TCRs, such as MAIT cells [273,274] and iNKT cells [275,276] (Figure 3). However, SAgs can also recognise γδ T cells, demonstrating that these toxins can bind TCRs lacking TCR α- or β-chains, and such interaction by certain SAgs involves TRGV9^+^ γδ TCRs [277,278,279,280]. MAIT cells typically encode TCRs with biased TRBV20-1 and TRBV6-2/6-3 gene usage [192], exposing them to SAgs such as SEB, TSST-1, and SpeC, which was proven in recent studies [273,274]. Indeed, SAg-triggered MAIT cells produce substantially high levels of effector cytokines (e.g., IFNγ), through an IL-12/IL-18-dependent mechanism that does not involve MR1. Such hyperactivation induces MAIT cell anergy both in vitro and in vivo, in humanized mouse models, impeding their anti-bacterial capabilities [273,274]. Highly activated MAIT cells form major contributors of cytokines, produced in the acute phase of streptococcal TSS (STSS), despite the GAS species lacking de novo riboflavin synthesis, indicating that SAg-driven MAIT cell activation underlies the cytokine storm seen in STSS patients [274]. All of this demonstrates the considerable impact SAgs have in clinical settings and the direct effects they have on both conventional αβ T cells and unconventional αβ or γδ T cells. It highlights the breadth of T cell populations that are targets for these toxins, and further research will define if SAg-targeting of T cells is more intricate than initially thought.

## 4. Conclusions

Cellular components of the adaptive immune system, notably T cells, evolved a multitude of ways to detect and respond to pathogenic bacterial infections, utilizing MHC and MHC-like molecules to present structurally and chemically different antigens from bacteria. MHC-restricted αβ CD4^+^ and CD8^+^ T cells, provide immune defense in response to peptidic antigens, whilst unconventional αβ (MAIT cells, CD1-restricted T cells) and γδ T cells, utilize MR1, CD1 molecules and BTN/BTNL molecules to respond to lipidic, metabolic, or undiscovered antigens of bacterial origin. Seminal research, notably in the past decade, helped to unravel how key antimicrobial T cell populations, notably unconventional subsets, recognize bacteria and the intrathymic developmental processes that are involved to seed populations into the periphery. In the case of γδ T cells, the process of identifying γδ T cell antigens is complicated and, as such, the nature of bonafide antigens that respond to γδ TCRs is largely unclear [26]. New developments shed light on γδ TCR-driven antigen recognition and further work could theoretically uncover novel antigens from bacterial sources. Similarly, the repertoire of MAIT cell antigens is expanding and the flexibility observed in the MR1-binding cleft suggest that many more antigens could be presented to MAIT cells or T cells that can react to MR1-restricted antigens. The next ten years will inevitably provide new discoveries that will help to further define the fundamental process underlying T cell control of bacterial infections. Such information will be incredibly informative for novel therapies and vaccination strategies. However, T cell immunity is not always sterilizing and bacteria have evolved ways to evade T cell immunosurveillance, utilizing strategies to hide in T cells or APCs or approaches that inhibit or exhaust T cell function. Furthermore, bacteria can release pore-forming toxins that kill T cells and potent exotoxins (SAgs) that bind TCRs, re-wiring the cells into a hyperactive state that induces anergy and the production of a cytokine storm that could be lethal to the host. Subsequently, strategies to vaccinate against bacterial toxins are equally as important as those that elicit protective, antimicrobial immunity. The production of SAgs is strongly associated with many life-threatening conditions, such as pneumonia, sepsis, and TSS, and these toxins accordingly have become pivotal targets for vaccination strategies in recent clinical trials. As such, it is important to learn more about how bacteria evade the adaptive immune system but also the breadth of interactions that bacterial toxins have evolved, in order to design therapies that neutralize their activity.

## Figures and Tables

**Figure 1 ijms-21-06144-f001:**
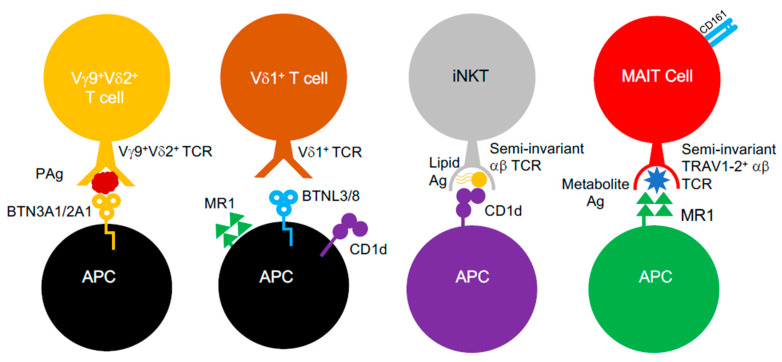
Unconventional αβ and γδ T cells. Unconventional αβ and γδ T cells populations and known interactions between their TCRs, the antigens they target and the MHC-like molecules involved.

**Figure 2 ijms-21-06144-f002:**
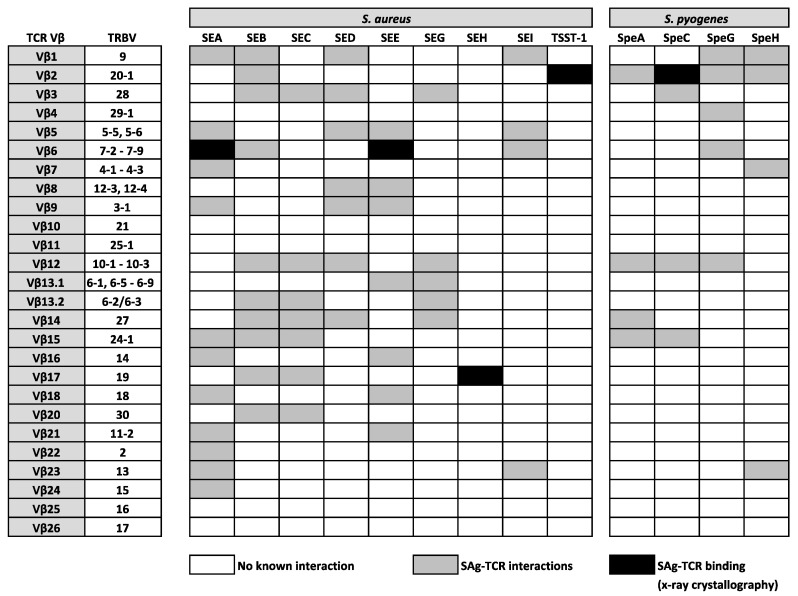
Established interactions between well characterised SAgs from *S. aureus* and *S. pyogenes* and human TCR β-chain variable (TRBV) regions.

**Figure 3 ijms-21-06144-f003:**
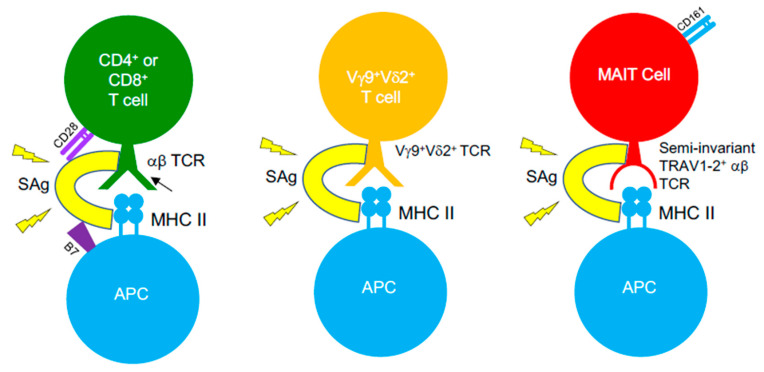
SAgs target αβ and γδ TCRs on different T cell subsets. SAgs bypass conventional TCR–pMHC interactions by binding TCRs from αβ and γδ T cells, outside the peptide-binding groove and cross-linking them to MHC class II molecules. Such an interaction with certain T cell subsets also involves binding interactions with co-stimulatory molecules on T cells (CD28) and their ligands on APCs (B7-2), indicating that SAgs are capable of forming quaternary complexes between T cells and APCs, in order to induce T cell hyperactivation.

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
