# Peer review of "T Cell Immunity to Bacterial Pathogens: Mechanisms of Immune Control and Bacterial Evasion"

_ijms, 2020, doi:10.3390/ijms21176144_

Round 1

Reviewer 1 Report

In the review titled, “T cell immunity to bacterial pathogens: mechanisms of immune control and bacterial evasion,” Shepherd and Mclaren provide a comprehensive overview of the role of T-cells in host response to bacterial infections. The review is well suited for publication in IJMS. However, a few changes will improve the overall accuracy of the review.

Major points:

  1. In the discussion regarding the role of T3SS, the authors suggest that CASP1 activation leads to apoptosis and the following sentence suggest that this apoptosis leads to secretion of pro-inflammatory cytokines. The papers describing CASP1 dependent apoptosis were published before the discovery of pyroptosis. Also, CASP1 activation could be protective in this context as demonstrated by a recent study (https://www.biorxiv.org/content/10.1101/2020.05.16.099929v1.full)

Minor points:

  1. The first sentences of the abstract and introduction are almost identical. It might be better to rephrase one of them.

  2. In the introduction lines 33-34: the sentence suggests that innate immune system does not have any form of memory. This sentence needs to be rephrased. Although, classically this was the view, recent studies have highlighted the presence of non-specific trained immunity. Trained immunity allows the innate immune system to respond more rapidly and robustly to repeat pathogenic challenges.

Author Response

In the review titled, “T cell immunity to bacterial pathogens: mechanisms of immune control and bacterial evasion,” Shepherd and Mclaren provide a comprehensive overview of the role of T-cells in host response to bacterial infections. The review is well suited for publication in IJMS. However, a few changes will improve the overall accuracy of the review.

Major points:

In the discussion regarding the role of T3SS, the authors suggest that CASP1 activation leads to apoptosis and the following sentence suggest that this apoptosis leads to secretion of pro-inflammatory cytokines. The papers describing CASP1 dependent apoptosis were published before the discovery of pyroptosis. Also, CASP1 activation could be protective in this context as demonstrated by a recent study (https://www.biorxiv.org/content/10.1101/2020.05.16.099929v1.full)

We would like to thank the reviewer for their critique here and for indicating the absence of reference to pyroptosis in our review. To address, we have re-worded the section of the review in question and have added extra wording to reflect how Shigella species utilise the inflammasome to induce pyroptosis (lines 539-544). We hope that this re- written section will appease the views of the reviewer.

Minor points:

The first sentences of the abstract and introduction are almost identical. It might be better to rephrase one of them.

We would like to thank the reviewer for their observation regarding the repetition in the first line of the abstract and introduction. To address this, we have rephrased the first sentence of the abstract (lines 9 and 10) so that it does not replicate the opening line of the introduction.

In the introduction lines 33-34: the sentence suggests that innate immune system does not have any form of memory. This sentence needs to be rephrased. Although, classically this was the view, recent studies have highlighted the presence of non-specific trained immunity. Trained immunity allows the innate immune system to respond more rapidly and robustly to repeat pathogenic challenges.

We would like to thank the reviewer for critique and making a profound point that cells of the innate immune system have been shown to develop memory, more known as “trained immunity”. To address this, we have re-written the sentence indicated above and include new wording in lines 32-35 which refer to innate immune cells possessing trained immunity.

Reviewer 2 Report

This is a comprehensive and balanced review about T cell responses against bacteria and bacteria mechanism to evade immune system action. The different T cell populations are well described. The review shows the latest advances in the knowledge of the processes by which T cells fight bacteria, playing special attention not only to the most known T cell subsets but also to the more recent discovered and more unknown T cell subtypes. The manuscript also incorporates interesting information about the action of memory T cells in response to bacterial infection.  Finally, the authors describe the strategies of bacterial immune evasion. The review is very well written in English, easy to capture the major points, well organized and the information is up to date. Overall, this is a well-written review in a topic with not many review papers and thus will be a very useful review article for the field of immunology.

Minor point

Please correct the format of the letters in the title

Author Response

This is a comprehensive and balanced review about T cell responses against bacteria and bacteria mechanism to evade immune system action. The different T cell populations are well described. The review shows the latest advances in the knowledge of the processes by which T cells fight bacteria, playing special attention not only to the most known T cell subsets but also to the more recent discovered and more unknown T cell subtypes. The manuscript also incorporates interesting information about the action of memory T cells in response to bacterial infection. Finally, the authors describe the strategies of bacterial immune evasion. The review is very well written in English, easy to capture the major points, well organized and the information is up to date. Overall, this is a well-written review in a topic with not many review papers and thus will be a very useful review article for the field of immunology.

Minor point

Please correct the format of the letters in the title

We thank the reviewer for spotting this error which, we believe, occurred during reformatting of the manuscript at the journal. This error has been corrected accordingly.